# Proteomic Study of Response to Copper, Cadmium, and Chrome Ion Stress in *Yarrowia lipolytica* Strains Isolated from Andean Mine Tailings in Peru

**DOI:** 10.3390/microorganisms10102002

**Published:** 2022-10-11

**Authors:** Tito Sánchez-Rojas, Abraham Espinoza-Culupú, Pablo Ramírez, Leo Kei Iwai, Fabio Montoni, Diego Macedo-Prada, Marcos Sulca-López, Yerson Durán, Mariella Farfán-López, Jennifer Herencia

**Affiliations:** 1Laboratory of Environmental Microbiology and Biotechnology, Faculty of Biological Sciences, Universidad Nacional Mayor de San Marcos, Lima 15081, Peru; 2Laboratory Research on Health Science, Universidad Señor de Sipán, Chiclayo 14001, Peru; 3Molecular Microbiology and Biotechnology Laboratory, Faculty of Biological Sciences, Universidad Nacional Mayor de San Marcos, Lima 15081, Peru; 4Laboratory for Applied Toxinology Center of Toxins, Immune-Response and Cell Signaling (LETA/CeTICS), Butantan Institute, São Paulo 05503-900, Brazil

**Keywords:** *Yarrowia lipolytica*, heavy metal ions, Andean mining, oxidative stress, proteomics

## Abstract

Mine tailings are produced by mining activities and contain diverse heavy metal ions, which cause environmental problems and have negative impacts on ecosystems. Different microorganisms, including yeasts, play important roles in the absorption and/or adsorption of these heavy metal ions. This work aimed to analyze proteins synthesized by the yeast *Yarrowia lipolytica* AMJ6 (Yl-AMJ6), isolated from Andean mine tailings in Peru and subjected to stress conditions with common heavy metal ions. Yeast strains were isolated from high Andean water samples impacted by mine tailings from Yanamate (Pasco, Peru). Among all the isolated yeasts, the Yl-AMJ6 strain presented LC50 values of 1.06 mM, 1.42 mM, and 0.49 mM for the Cr^+6^, Cu^+2^, and Cd^+2^ ions, respectively. Proteomic analysis of theYl-AMJ6 strain under heavy metal stress showed that several proteins were up- or downregulated. Biological and functional analysis of these proteins showed that they were involved in the metabolism of proteins, nucleic acids, and carbohydrates; response to oxidative stress and protein folding; ATP synthesis and ion transport; membrane and cell wall; and cell division. The most prominent proteins that presented the greatest changes were related to the oxidative stress response and carbohydrate metabolism, suggesting the existence of a defense mechanism in these yeasts to resist the impact of environmental contamination by heavy metal ions.

## 1. Introduction

In Peru, mining and metallurgical activity dates back to long before the pre-Incan era, and the oldest gold gadgets found date back to approximately 4000 years ago [1]. Currently, mining is the predominant sector of the Peruvian economy and has allowed Peru to become one of the main mineral producers in the world. As a negative environmental impact of the extraction process, the tailings and acidic waters rich in heavy metals produced by this process pollute the environment, especially in areas surrounding mining facilities [2]. High concentrations of essential and nonessential metal ions are toxic to the environment, especially when their concentrations exceed the maximum permissible limits according to environmental quality standards [3]. The amounts of heavy metal ions present in mine tailings have increased in recent decades [4,5,6]. These polluting heavy metals are discarded into the environment, entering rivers, lakes, and other bodies of water where there is anthropological activity, such as industrial and mining activity. The elimination of these wastes containing heavy metals is a serious environmental concern with consequences for public health, since these metals are toxic and can be lethal to humans [7,8].

Heavy metal ions found in mine tailings, such as cadmium, copper, chromium, lead, and mercury ions, are toxic to organisms even at low concentrations [9]. However, certain microorganisms, such as bacteria and yeasts isolated from areas contaminated by heavy metals, have great potential for adaptation to these metal ions via various resistance mechanisms, such as activation of the redox antioxidant system (AORS) to reduce reactive oxygen species (ROS) and chelation of metal ions by phytochelatins via glutathione (GSH) and metallothionein (MT) [10,11,12].

Among these organisms, the extremophilic yeast *Yarrowia lipolytica* AMJ6 (Yl-AMJ6), with resistance to high concentrations of heavy metal ions, was reported for the first time by our research group after being isolated from water samples collected from highland Andean mine tailings in Peru. To date, few studies have examined the resistance to heavy metal ions of this species of yeast and, to the best of our knowledge, there have been no studies describing the proteomic modulation that occurs in this yeast species under heavy metal treatment. Therefore, in the present work we studied the proteomic profile of the Yl-AMJ6 strain with resistance to heavy metal ions, when it was cultivated under Cd^+2^, Cu^+2^, and Cr^+6^ stress.

## 2. Materials and Methods

### 2.1. Reagents

Taq polymerase, dNTP mix, ultrapure water, loading buffer, and Tris-EDTA buffer were purchased from Merck (Darmstadt, Germany); rDNA and LSU primers were purchased from Macrogen Korea; and protease inhibitor cocktail IV, agarose, chloroform, isoamyl alcohol, protease inhibitor cocktail, acrylamide, and methanol were purchased from Sigma Chemical Co., Ltd. (St. Louis, MO, USA).

### 2.2. Sampling and Collection

The samples were collected in months of the dry and rainy seasons (June and November, respectively) in 2015 from different water bodies in the high Andes, at more than 4000 m above sea level (m.a.s.l.). These water bodies are located in the headwaters of the Mantaro River basin, between the Junín and Pasco departments in the Yanamate and Chinchaycocha Lakes in the Peruvian central highlands. Seven sampling stations, identified as M1 to M7, were chosen and georeferenced through the use of a GPS device (GARMIN eTrex 30). Duplicate water samples (1000 mL) were collected in sterile glass bottles for microbial isolation. Samples were collected at each station following the standard procedures for water and wastewater evaluation [13]. The bottles containing the water samples were labeled, transported, and stored at 4 °C. The coordinates and some physicochemical characteristics of the water samples are described in Table 1.

### 2.3. Water Physicochemical Analysis and Heavy Metal Measurements

The temperature, electric conductivity, and total dissolved solids content were measured in situ at each station with a calibrated EXTECH-DO700 (Extech, Nashua, NH, USA) multiparameter meter. Water samples were collected in bottles with 20 drops of nitric acid at 10% per liter, as described in Rice’s [13] standard methods for the examination of water and wastewater. The samples were transported on ice for heavy metal quantification, which was carried out through a service performed by the Corrosion and Protection Institute of the Pontificia Universidad Católica del Peru (Table 2). The protocols for sampling were as described in the guidelines from ANA (Autoridad Nacional del Agua—Peru) [14].

### 2.4. Yeast Isolation and Culture Conditions

Ten milliliter measures of the collected water samples were inoculated in 250 mL flasks containing 90 mL of YPG broth (5 g yeast extract, 10 g peptone, and 30 g glucose in 1000 mL of distilled water) at pH 4.0 and incubated at 20 °C under constant stirring at 150 rpm for 1 week. Culture that exhibited turbidity were collected, seeded in YPG agar, and incubated at the same temperature for the same duration. After the incubation period, yeast colonies were isolated according to their culture characteristics, such as the color, border shape, and size of colonies, following the methodology proposed by Acosta [15]. Some yeast strains were chosen for further testing of their resistance to various heavy metals. Selected yeast colonies were cryopreserved at −20 °C for subsequent evaluation.

### 2.5. Heavy Metal Tolerance Determination

The metal salts K_2_Cr_2_O_7_, CdCl_2_, and CuSO_4_ were used in this experiment. Stock solutions of each metal salt were prepared to determine the minimum inhibitory concentration (MIC) [16]. The yeast strains were inoculated in YPG broth supplemented with different concentrations of heavy metal ions (from 0.1 to 2 mM) and incubated at 20 °C for three weeks, with constant stirring at 150 rpm. The strains that grew under high concentrations of heavy metals were selected and cryopreserved at −20 °C in semisolid YPG medium supplemented with 10% glycerol. All assays were performed in triplicate.

### 2.6. Molecular Identification and Phylogenetic Analysis of Selected Yeasts

DNA extraction was performed using exponential phase cultures, and the extracted DNA was used as a template to amplify the D1/D2 domains of the rDNA LSU with the primers F63 (5′-GCATATCAATAAGCGGAGGAAAAG-3′) and LR3 (5′-GGTCCGTGTTTCAAGACGG-3′) [17]. The PCR products were subjected to 2% agarose gel electrophoresis in TAE buffer. The gels were then stained with ethidium bromide and photographed under UV light. The amplified products were sequenced by Molecular Cloning Laboratories (MCLAB, San Francisco, CA, USA)

In silico analysis of nucleotide sequences was performed using bioinformatics tools. BioEdit [18] was used to evaluate and obtain consensus sequences from chromatograms. The nucleotide sequences obtained were aligned with other nucleotide sequences stored in GenBank using Blastn [19] for identification. Subsequently, ClustalW [20] was used for multiple-sequence alignments and to perform phylogenetic analysis of the selected yeasts. The nucleotide sequences obtained were deposited in the GenBank database. The phylogenetic analysis was performed using the MEGA v10 program and the best evolutionary model (from the best DNA model analysis) was determined using the neighbor-joining method and the nucleotide-substitution Kimura two-parameter distance model [21].

### 2.7. Growth Kinetics and Median Lethal Concentration (LC_50_) Assay

Inoculum of 30 mL of each strain was transferred to flasks containing 300 mL of YPG broth and incubated at 20 °C with constant stirring at 150 rpm. Optical density measurements were used to observe the growth of the strains by collecting a sample of 2 mL of the culture every 6 h from the beginning to the end of the assay for 72 h [22]. Each inoculum collected was diluted (10^−1^ to 10^−5^), and 100 µL of the last dilution was cultured in YPG agar. These cultures were incubated at 20 °C for 48 h to calculate the CFU/mL value and determine the cell viability and growth kinetics of the yeasts. The median lethal concentration (LC_50_) was determined by exposing the AMJ3 and AMJ6 strains (2.5 × 10^8^ CFU/mL and 5.0 × 10^8^ CFU/mL for, respectively) to different concentrations of each heavy metal ion for 3 h. The LC_50_ values were calculated from cell viability of the yeasts based on the number of CFUs in plates containing YPG agar compared with the control without treatment. All assays were performed in triplicate.

### 2.8. Heavy Metal Biosorption

The growth kinetics and LC_50_ tests were used as references to measure the biosorption of Cr^+6^ and Cu^+2^ ions by the AMJ3 and AMJ6 strains. The concentrations of these ionic metals were as follows: 80 ppm (1.26 mM) of Cu^+2^ ions for both strains and 15 ppm (0.145 mM) and 50 ppm (0.49 mM) of Cr^6+^ ions for the AMJ3 and AMJ6 strains, respectively. All sets of experiments were performed in a fixed volume of 250 mL of YPG broth with a single metal ion solution in a 1 L Erlenmeyer flask. The fungal biomass (10^7^ CFU/mL) was exposed to the metal solutions for 96 h at 20 °C on an orbital shaking incubator at 150 rpm. Fifty milliliters of each culture was taken every 24 h, the biomass was separated by centrifugation at 4500 rpm for 20 min, and the supernatant was analyzed for the residual metal concentration using the plasma emission spectroscopy method. This experiment was performed in triplicate.

### 2.9. Protein Extraction

The Yl-AMJ6 strain was cultured in the presence or absence of heavy metals: Cu^+2^ (0.50 mM) for 48 h or Cd^+2^ (0.11 mM) and Cr^+6^ (0.19 mM) for 72 h at 25 °C and 150 rpm in 350 mL of Davis minimal medium (ammonium sulfate (1 g/L), dextrose (1 g/L), dipotassium phosphate (7 g/L), magnesium sulfate (0.1 g/L), monopotassium phosphate (2 g/L), and sodium citrate (0.5 g/L); pH (4.0 ± 0.2)) supplemented with malt extract (1%) and yeast extract (1%). A total of 250 mL was collected from the culture in the exponential phase and transferred to a preweighed 50 mL conical tube, followed by centrifugation at 7500 rpm for 10 min at 4 °C. The supernatant was discarded to obtain the pellet which was further weighed and dried. Each dry pellet was washed twice using 100 mM and 10 mM Tris-HCl buffer (pH 8.8) and then resuspended in 10 mM Tris-HCl buffer (2 mL/g pellet) (pH 8.8) supplemented with protease inhibitor cocktail IV (Calbiochem, Merck, Feltham, UK). The resuspended pellets were sonicated using a Misonix Sonicator, model XL-2000-010 (ALT, San Diego, CA, USA). Sonication of the cell suspensions was performed at 20 kHz for 40 s (20 cycles) with intervals of 20 min for cooling between each pulse. Finally, the cell lysate was centrifuged at 14,000 rpm and 4 °C, and the supernatant was collected and stored at −20 °C. The protein content in the supernatant was quantified using Bradford reagent. Equal amounts of protein were loaded into each lane on an SDS-PAGE gel for electrophoresis analysis [23]. All experiments were performed in triplicate.

### 2.10. Digestion and Tagging of Proteins with iTRAQ-8-plex^®^ Reagent

Digestion and tagging of the proteins were performed by the Proteomic Laboratory at the National Center for Biotechnology (CNB), Madrid, Spain. To carry out proteomic studies, the total protein concentration was determined using a Pierce 660 nm Protein Assay Kit (Thermo Fisher Scientific, Rockford, IL, USA). For digestion, 50 µg of protein from each treatment was precipitated using the methanol/chloroform method [24]. The supernatant was carefully removed and protein pellets were resuspended and denatured in 20 µL of a solution containing 7 M urea, 2 M thiourea, and 0.1 M TEAB (triethylammonium bicarbonate), pH 7.5, (SERVA Electrophoresis GmbH, Heidelberg, Germany), reduced with 1 µL of 50 mM Tris(2-carboxyethyl) phosphine (TCEP, AB SCIEX Foster City, CA, USA) (pH 8.0) at 37 °C for 60 min, followed by 2 µL of 200 mM cysteine-blocking reagent methyl methanethiosulfonate (Thermo Fisher Scientific, Rockford, IL, USA) for 10 min at room temperature. Samples were diluted with 50 mM TEAB to a volume of 120 µL to reduce the urea/thiourea concentration. Digestion was initiated by adding 2 µg of sequencing-grade modified trypsin (Sigma Aldrich, St. Louis, MO, USA) to each sample at a ratio of 1:25, and the samples were then incubated at 37 °C overnight on a shaker. The solvent from the digested samples was evaporated to dryness in a vacuum concentrator (Hetovac VR-1, Heto Lab Equipment, Lillerød, Denmark).

Each trypsin-digested sample was reconstituted with 80 µL of 70% ethanol/50 mM TEAB and labeled at room temperature for 2 h with a half unit of iTRAQ 8-plex reagent (AB SCIEX, Foster City, CA, USA). The iTRAQ labeling was performed according to the following schema: iTRAQ 113 reagent = control 1; iTRAQ 114 reagent: control 2; iTRAQ 115 reagent: Cd^2+^ replicate 1 (Cd1); iTRAQ 116 reagent: Cd^2+^ replicate 2 (Cd2); iTRAQ 117: Cu^2+^ replicate 1 (Cu1); iTRAQ 118 reagent: Cu^2+^ replicate 2 (Cu2); iTRAQ 119 reagent: Cr^2+^ replicate 1 (Cr1); iTRAQ 120 reagent: Cr^2+^ replicate 2 (Cr2). After labeling, the samples were mixed and the labeling reaction was stopped by evaporation in a Speed Vac. Salts were washed using a reversed-phase Sep-Pak C18 cartridge (Waters, Milford, MA, USA) according to the manufacturer’s recommendations.

### 2.11. Mass Spectrometry

Five microliters of sample (0.4 µg/µL) was injected into a 2D nano liquid chromatography (nanoLC) ESI-MS/MS system (Eksigent Technologies nanoLC Ultra 1D Plus, AB SCIEX, Foster City, CA, USA) coupled to a High Speed TripleTOF 5600 Mass Spectrometer (SCIEX, Foster City, CA, USA) with a Nanospray III source. The trap column was an Acclaim PepMap 100 (ThermoFisher Scientific, Rockford, IL, USA) with a 5 µm particle diameter and 100 Å pore size, switched online with the analytical column. The loading pump delivered a solution of 0.1% formic acid in water at 2 µL/min. The nanopump provided a flow rate of 300 nL/min and was operated under gradient elution conditions, using 0.1% formic acid in water as mobile phase A and 0.1% formic acid in acetonitrile as mobile phase B. These assays were performed in the Proteomic Laboratory at the National Center for Biotechnology (CNB), Madrid, Spain.

### 2.12. Data Analysis and Database Searching

MS/MS spectra were exported to mgf format using PEAKS Studio version X software (Bioinformatics Solutions Inc., Toronto, ON, Canada) and searched using Mascot (Matrix Science, version 2.4.0, Boston, MA, USA), X! Tandem (version 2013.02.01.1) and Myrimatch (version 2.2.140) against a composite target/decoy database built from the 6454 *Y. lipolytica* (strain CLIB 122/E 150) reference proteome sequences at UniProt Knowledgebase [25] (October 2020), together with commonly occurring contaminants. After recalibration of patent ion mass measurements using high-scoring X! Tandem hits, search engines were configured to match potential peptide candidates with a mass error tolerance of 10 mg/g and fragment ion tolerance of 0.02 Da, allowing for up to two missed tryptic cleavage sites and an isotope error (^13^C) of 1. In addition, fixed MMTS modification of cysteine and variable oxidation of methionine, pyroglutamic acid from glutamine or glutamic acid at the peptide N-terminus, acetylation of the protein N-terminus, and modification of lysine, tyrosine, and the peptide N-terminus with iTRAQ 8-plex reagents were considered. Score distribution models were used to compute peptide-spectrum match *p* values of 1, and spectra recovered by an FDR < 0.01 (peptide-level) filter were selected for quantitative analysis. Approximately 5% of the signals with the lowest quality were removed prior to further analysis. Differential regulation was measured using linear models, and statistical significance was measured using q values (FDR). All results for the treatments with the different metals were normalized to those for control without treatment (Appendix A). All analyses were conducted using software from Proteobotics of CNB [26].

### 2.13. Network Analysis

The proteins identified in the previous tests that were overexpressed in the presence of Cu^+2^, Cd^+2^, and Cr^+6^ ions were analyzed against the STRING database [27] to predict protein interactions using the *Y. lipolytica* strain. The network parameters were obtained by default and with a score of 0.7 (high confidence). Graphs were obtained to depict the interactions of proteins, represented by nodes.

## 3. Results

### 3.1. Study Area and Physicochemical Parameters

Sampling station georeferencing and the corresponding physicochemical parameters are described in Table 1, and the results of the quantification of heavy metals in the water samples are shown in Table 2.

### 3.2. Isolation and Molecular Characterization

A total of 10 yeast strains were isolated under laboratory conditions, among which isolates of the genus *Yarrowia* presented better growth and tolerance to heavy metal ion treatment than the other isolated strains. Therefore, we decided to perform additional experiments with these isolates. The AMJ3 and AMJ6 strains were chosen because they showed a high resistance profile, and both strains were isolated from water samples from Lake Junín. Heavy metal ion resistance was verified by analysis of the concentrations of heavy metals in water samples from which these two strains were isolated (Figure 1A,B). These results confirmed that the concentrations of various heavy metals were high according to the Environmental Quality Standards (ECA, 2017) for Water (SD No. 004-2017-MINAM, Peru).

The nucleotide sequences of *Yarrowia* strains were grouped in a cluster that was phylogenetically related to *Yarrowia lipolytica* (Figure 1C). The Blastn results confirmed a 100% identity with the nucleotide sequences of *Y. lipolytica* present in the database. The accession numbers for these sequences were MH656398 and MH656399 for *Y. lipolytica* AMJ6 and *Y. lipolytica* AMJ3 strains, respectively.

### 3.3. Growth and Cytotoxicity Studies against Heavy Metal Ions

The two *Y. lipolytica* strains (AMJ6 and AMJ3) showed similar growth curves. The exponential growth phase started at 30 h after culture initiation and continued until approximately 44 h. This was followed by a stationary phase that lasted approximately 36 h and 24 h for the AMJ3 and AMJ6 strains, respectively (Figure 2). These curves were similar to the growth curves of marine *Y. lipolytica* strains, in which the exponential phase begins at 36 h [22]. However, a distinct growth pattern was observed when the strains were exposed to different metal concentrations (Appendix A). The evaluation of the kinetics and growth patterns were relevant for the subsequent LC_50_ and proteomic analyses.

For LC_50_ determination, the *Y. lipolytica* AMJ6 strain was chosen because it was the most resistant to heavy metals, and the results indicated that 50% (shown in Figure 3 by the dashed line) of the yeast population died when exposed to the concentrations of metals mentioned for 3 h (Figure 3). The approximate LC_50_ was 1.06 mM for Cr^+6^, 1.42 mM for Cu^+2^, and 0.49 mM for Cd^+2^. The LC_50_ results showed that Cr^+6^ and Cd^+2^ ions were the most lethal, followed by Cu^+2^ ions. At other concentrations, the percentages of viable cells with 0.96 mM Cr^+6^, 1.57 mM Cu^+2^, and 0.44 mM Cd^+2^ were approximately 100, 30, and 60%, respectively.

### 3.4. Biosorption Profile

The AMJ3 and AMJ6 strains presented high percentages of biosorption of Cu^2+^ ions. The biosorption percentages were 91.9% and 90.3% for strains AMJ6 and AMJ3, respectively, between 48 and 72 h of cultivation. The biosorption by strain AMJ6 was slightly higher than that by strain AMJ3 (Figure 4A). With regard to the biosorption of Cr^+6^ ions, the AMJ3 and AMJ6 strains both presented biosorption at 72 h, with an average value of 90% (Figure 4B).

### 3.5. Effect of Heavy Metal Ions on the Protein Profile

SDS-PAGE was performed to evaluate the protein profiles of the yeast strains exposed to the metal ions. We observed differences in the profiles compared to that of the control without treatment, as shown in Figure 5.

### 3.6. Identification of Proteins Present in Y. lipolytica under Treatment with Heavy Metal Ions

Mass spectrometry analysis of *Y. lipolytica* cultured in the presence of heavy metal ions allowed the identification of 760 proteins. Among them, the differentially expressed proteins of the yeast in the presence of Cr^+6^, Cd^+2^, and Cu^+2^ were identified. Under treatment with Cr^+6^ ions, 68 proteins were upregulated and 50 were downregulated. In the presence of Cd^+2^ ions, 128 proteins were upregulated and 124 were downregulated. Under treatment with Cu^+2^ ions, 208 proteins were upregulated and 182 were downregulated. All these results were obtained using a control treatment without heavy metal ions (Figure 6).

### 3.7. Proteins Regulated in Y. lipolytica AMJ6 in the Presence of Cr^+6^

The results showed that treatment with Cr^+6^ led to differential regulation of 118 proteins (up- and downregulated) compared to untreated control (Appendix A). The biological function classification of these proteins showed that 23.53% of the upregulated proteins and 10% of the downregulated proteins participated in protein metabolism. In addition, 13.24% of the upregulated proteins and 18% of the downregulated proteins were related to carbohydrate metabolism. Additionally, 11.76% of the upregulated proteins and 8% of the downregulated proteins were related to the oxidative stress response (Figure 7). In the untreated control, 564 proteins that were not differentially regulated were observed (data not shown).

### 3.8. Proteins Regulated in Y. lipolytica AMJ6 in the Presence of Cd^+2^

The *Y. lipolytica* culture under treatment with Cd^+2^ showed 252 differentially regulated proteins (up and downregulated) compared to the untreated control (Appendix A). The biological function classification of these proteins showed a higher percentage (21.1%) of upregulated proteins involved in protein metabolism than downregulated proteins (12.1%; Figure 8). Furthermore, 12.5% of the upregulated and 14.52% of the downregulated proteins were related to carbohydrate metabolism, and 10.16% of the upregulated and 6.45% of the downregulated proteins were associated with oxidative stress. The proteins expressed in the untreated control included 384 proteins that were not differentially regulated (data not shown).

### 3.9. Proteins Regulated in Y. lipolytica AMJ6 in the Presence of Cu^+2^

The *Y. lipolytica* culture under treatment with Cu^+**2**^ showed 390 differentially regulated proteins (up- and downregulated) compared to the untreated control (Appendix A). The biological function classification of these proteins showed that 31.25% of the upregulated and 16.48% of the downregulated proteins were related to protein metabolism (Figure 9). Furthermore, 4.81% of the upregulated and 7.69% of the downregulated proteins were related to the oxidative stress response, and 4.81% of the upregulated and 7.69% of the downregulated proteins were associated with carbohydrate metabolism. Protein expression analysis of the untreated control showed 207 proteins that were not differentially regulated (data not shown).

### 3.10. Common and Unique Proteins Identified under Treatment with Cr^+6^, Cd^+2^, and Cu^+2^ Ions

Analysis of the all the data using BioVenn (web application) [28] showed common protein expression when *Y. lipolytica* AMJ6 was cultured with Cr^+6^, Cd^+2^, and Cu^+2^ (Figure 10). Among the upregulated, proteins, chromium, cadmium, and copper induced the differential expression of 20, 47 and 112 unique proteins, respectively. Furthermore, all the heavy metal ions induced upregulation of 17 common proteins (Appendix A). Furthermore, among the downregulated proteins, the differential expression of 6, 35, and 84 exclusive unique proteins was induced by chromium, cadmium, and copper, respectively. Likewise, all the heavy metal ions induced downregulated expression of 31 common proteins (Appendix A).

### 3.11. Protein Interaction Network

The interactions among the 68 proteins upregulated under chromium treatment (0.19 mM) are represented in Figure 11, with 12 nodes and an average clustering coefficient of 0.572. The main biological process identified was cell redox homeostasis among four proteins: thiol-specific peroxidase (Q6CGH1), peroxiredoxin (Q6C2G0), thioredoxin-like protein (Q6CEJ7), and protein disulfide-isomerase (Q6C781). This was followed by the cellular response to toxic substances, identified among six proteins, with three unique proteins: glutathione peroxidase (Q6C7A8), catalase (Q6C3H7), and thioredoxin reductase (TRP1).

The protein interaction network with 128 proteins upregulated under cadmium treatment (0.11 mM) is represented in Figure 12 with eight nodes and an average clustering coefficient of 0.917. The main biological process identified was cell oxidative stress among five proteins: glutathione reductase (Q6C5H4), peroxiredoxin (Q6C2G0), thioredoxin (Q6C399), superoxide dismutase (Q6C662), and thioredoxin reductase (Q6C7L4). This was followed by cellular redox homeostasis, with all the same proteins except two: thiol-specific peroxidase (Q6CEJ7) and dihydrolipoyl dehydrogenase (Q6C8C6).

Protein interactions identified under copper treatment (1.26 mM) are represented in Figure 13, with 208 nodes and an average clustering coefficient of 0.464. The main biological processes identified were response to oxidative stress and cellular response to chemical stimulus among eight proteins: thioredoxin reductase (Q6C7L4), Q6CCY8, superoxide dismutase (Q6C662), B5FVB6, Q6CBD6, Q6CD63, thioredoxin-like protein (Q6CEJ7), and glutathione peroxidase (Q6C7A8). This were followed by the response to ROS, among three proteins with two unique proteins: superoxide dismutase (SOD1) and thioredoxin reductase (TRP1).

The protein interaction network of the 17 proteins upregulated under the three metal treatments is represented in Figure 14, with 17 nodes and an average clustering coefficient of 0.41. The main biological process identified was the response to oxidative stress and the cellular response to chemical stimulus. There were seven common proteins: glycoside hydrolase (Q6C6P1), an enzyme that participates in the synthesis of carbohydrates; the protein Q6CC94, which participates in the endocytosis of yeast and has one protein domain superfamily, AH/BAR, involved in the maturation of the vesicular Golgi complex; phosphoglycerate mutase (Q6CFX7), which functions as an isomerase via glycolysis; thiol-specific peroxidase (Q6CEJ7), which catalyzes the reduction of hydrogen peroxide and organic compounds; thioredoxin reductase (Q6C7L4), which participates in the formation of redox-active disulfide bonds; the protein Q6C177, which is related to NADPH-dependent oxidoreductase activity; and the protein Q6C1A8, which functions as a cofactor for the metallic ion Mn^+2^. To assess the variations in the expression of the 17 common proteins, we generated a heatmap, as shown in Figure 15, based on Log2 ratio, red (0.5), green (1.0), and blue (1.5). The distances between clades are Euclidean distances.

## 4. Discussion

Organisms such as plants, algae, bacteria, filamentous fungi, and yeasts are capable of removing heavy metals from different sources of water in the environment [29,30]. These fungi have shown tolerance to various heavy metals, including copper, chromium, lead, arsenic, cobalt, and cadmium. Similar results were found by Oladipo [31] in samples from mine tailings from Nigeria. They found various species of fungal strains such as *Fomitopsis meliae*, *Trichoderma ghanense*, and *Rhizopus microsporus*, all of which were tolerant to heavy metals such as cadmium, copper, lead, arsenic, and iron.

In this work, we studied the resistance/tolerance and synthesis of diverse proteins in strains of the yeast *Y. lipolytica* isolated from natural water bodies contaminated with mine tailings in Andean mining areas in Peru. Several concentrations of copper were tested against *Y. lipolytica* AMJ6 and the LC_50_ of copper was determined to be 1.42 mM. These yeast strains were isolated from water samples of environmental origin, and the resistance pattern was expected to be different compared to those of other yeast species [32]. The resistance patterns of *Saccharomyces cerevisiae* against copper showed an LC_50_ between 0.047 mM and 0.057 mM [33], with similar values obtained for a baker’s yeast strain, which presented an LC_50_ between 0.430 mM and 0.462 mM. In contrast, *Y. lipolytica* AMJ6 presented higher resistance against copper ions, as this yeast was isolated from water samples contaminated by mine tailings from the Yanamate mining area of the central highlands of Peru [34]. Yeasts isolated from samples of environmental origin tend to be more resistant to heavy metals, even more resistant than other microorganisms such as bacteria, due to their ability to adapt to different environmental conditions, for example, to extreme conditions of pH, temperature, and nutrient availability, as well as to high concentrations of heavy metals [35]. To date and to the best of our knowledge, no scientific research has been published on *Y. lipolytica* isolates from Andean mine tailings.

Our LC_50_ analysis results showed that Cr^+6^ ions presented higher toxicity for both the AMJ3 and AMJ6 strains (absence of protein in gel and cytotoxicity in comparison with other heavy metals), followed by Cd^+2^ and Cu^+2^, after the cells were exposed to the metal for a period of 3 h. The stress produced by these heavy metals can modify the morphological characteristics of yeasts [36]. Rao [37] reported structural changes in two marine strains of *Y. lipolytica* (NCIM 3589 and NCIM 3590), with it revealing different shapes compared to the normal ellipsoidal shape, such as rounded, oval, or elongated, under different heavy metal stresses. With regard to biofilm formation, *Y. lipolytica* isolated from seawater samples from Mumbai, India, presented resistance to Cr^+6^ (2 to 4 mM), Cd^+2^ (2 to 5 mM), and Cu^+2^ (10 to 25 mM), on YPG plates. Additional analysis with other heavy metals, such as Pb and Hg, may reveal different MIC profiles between yeasts from mine tailings and marine environments [22].

Certain microorganisms use metal ions as parts of their cellular metabolism; therefore, some of them, such as fungi, are used for the elimination of heavy metals present in the environment [38,39,40]. Fungi play a very important role in the ecosystem as efficient recyclers [41] and are also used in biotechnology to produce antibiotics and other metabolites for commercial use. *Y. lipolytica* has been widely studied and used in biotechnology due to its ability to secrete products such as lipases [42]. Under laboratory conditions, the *Y. lipolytica* AMJ6 strain showed growth similar to that of other strains of the same species isolated from marine environments [22], with an exponential phase of approximately 36 h.

In this work, the *Y. lipolytica* yeast strain AMJ6 was subjected to treatments with Cr^+6^, Cd^+2^, and Cu^+2^ ions and the LC_50_ values were determined (Figure 3), in order to identify the proteins synthesized in the presence of these metals. The *Y. lipolytica* strain isolated from seawater showed tolerance to heavy metals such as Pb^+2^, Cr^+6^, Zn^+2^, Cu^+2^, As^+5^, and Ni^+2^. As the concentrations of these metals increased, the growth rate of the yeast decreased [36]. The authors suggested that these metals influence the growth of yeasts, consistent with our data showing that higher doses of Cr^+6^ were lethal to *Y. lipolytica*, decreasing the population growth rate and increasing protein denaturation in the cultures. In addition, we observed that exposure to Cr^+6^ significantly changed the protein profile (Figure 5). These results suggest that treatment with Cr^+6^ was harmful to normal protein expression in *Y. lipolytica*.

Proteomic analysis showed diverse proteins that were differentially regulated under chromium treatment, with a total of 118 up- and downregulated proteins. Toxicity analysis of chromium with the sod1 Delta mutant strain of *S. cerevisiae* (deficient for Cu^+2^, Zn-superoxide dismutase) showed that this yeast was sensitive to Cr^+6^ when cultured aerobically, showing resistance to this ion under aerobic conditions. This sensitivity under aerobic conditions was due to the oxidative action of this ion on various types of glycolytic enzymes and heat shock proteins (HSPs) [43]. In our study, we observed downregulation of the protein superoxide dismutase (Q6CFA1), which has antioxidant activity, upon treatment with Cr^+6^, which would indicate that this metal usually causes oxidative damage in the cells, affecting the native structure of proteins, for example, mitochondrial enzymes and HSPs [44]. Our results also showed some mitochondrial proteins that were modulated upon chromium treatment, such as glycerol-3-phosphate dehydrogenase (Q6CEQ0), isocitrate dehydrogenase [NAD] (Q6CA33), aldehyde dehydrogenase (Q6C2W9), and HSPs. As already reported, Cr^+6^ can affect the structure and degradation of proteins, as well as DNA, lipids, and other supramolecular biological compounds due to the oxidizing effect that this metal has on ROS production [43].

STRING protein interaction analysis showed that glutathione metabolism involves spermidine synthase (Q6C3P7), an enzyme responsible for transforming putrescine into spermidine, a polyamine with antioxidant activity that is involved in the stabilization of negative charges of DNA, RNA transcription, and protein synthesis, as well as in the prevention of the generation of hydrogen peroxides and oxidation of secondary compounds due to its heavy-metal-chelating activity [45]. The role of spermidine on transport-related genes, and genes involved in methionine, arginine, lysine, NAD, and biotin biosynthesis has been demonstrated in *S. cerevisiae* [46]. Exogenous polyamines such as spermidine have been extensively in the context of heavy metal tolerance in different plant species [47]; therefore, upregulation of spermidine synthase under chromium treatment suggests a mechanism of resistance to heavy metals of the *Y. lipolytica* AMJ6 strain under study.

Cell redox homeostasis and the cellular response to toxic substances involve the upregulation of glutathione peroxidase (Q6C7A8), peroxiredoxin (Q6C2G0), and thioredoxin reductase (Q6C7L4), proteins involved in defense mechanisms that are widely distributed in different organisms in the cellular response to oxidative stress under heavy metal exposure. In previous studies of *Y. lipolytica* in the presence of copper and uranium, upregulation of these proteins, better known as superoxide dismutases, was reported [48]. In addition, regarding the relationships of these enzymes, it has been shown in *S. cerevisiae* that the activity of peroxiredoxin is dependent on thioredoxin reductase and glutathione [49]; therefore, our results suggest the existence of the same dependency in *Y. lipolytica*.

In a previous study, the fungi *Aspergillus versicolor*, *Aspergilus fumigatus*, *Paecilomyces* sp., *Trichoderma* sp., *Microsporum* sp., and *Cladosporium* sp. showed high tolerance to Cd^+2^ ions, with MICs between 1000 and 4000 mg/mL [50]. In *Y. lipolytica*, we observed upregulation of proteins such as malate dehydrogenase, glycoside hydrolase, phosphoglycerate mutase, and transaldolase, which participate in different biochemical reactions in yeast. Jacobson mentioned that in *S. cerevisiae*, Cd^+2^ poisoning is related to the misfolding and aggregation of cytosolic proteins, where various types of chaperones are directly affected [51]. Therefore, treatment with Cd^+2^ ions regulates the synthesis of these types of proteins, suggesting that this response is a type of defense against Cd^+2^ poisoning. Likewise, our results showed upregulation of the enzymes glutathione synthetase (Q6CBL1) and glutathione reductase (Q6C5H4), which participate in the metabolism of glutathione. A similar result was obtained by Vido and coworkers using *S. cerevisiae* [52]. We also observed the synthesis of certain proteins that act as antioxidants in *Y. lipolytica* AMJ6 in the presence of Cd^+2^, such as thioredoxins (Q6C399) [52], suggesting that redox signaling mediated by the thioredoxin and glutathione systems is important for cellular defense against Cd^+2^ ions. Therefore, we hypothesize that total protein synthesis can be affected even by low concentrations of Cd^+2^ ions, and may further affect the endoplasmic reticulum, according to studies performed in *S. cerevisiae*, an effect that manifested as the misfolding of various proteins [53,54].

Experiments in culture medium supplemented with Cu^+2^ ions showed that treatment of *S. cerevisiae* with Cu^+2^ in YPG and YPD media increased the shelf life of the yeast and decreased the generation of superoxides [55]. In contrast, our results showed that copper supplementation was not harmful to *Y. lipolytica*. In addition, of the 390 identified proteins, those related to oxidative stress represented 4.8% and 7.7% of the proteins that were up- and downregulated, respectively (Figure 10, Appendix A). Furthermore, treatment with Cu^+2^ preferentially upregulated proteins related to protein metabolism, ATP synthesis, and ion transport, and downregulated proteins related to proteins and carbohydrate metabolism. Compounds that contain Cu^+2^ in their chemical structure are currently used as antimicrobial agents due to the reducing capacity of copper, which can reduce oxygen to superoxide or hydrogen peroxide, both of which are ROS [56,57].

Proteomic analysis of the AMJ6 strain in the presence of Cu^+2^ showed upregulated proteins with functions were related to oxidative stress and oxidoreductive activity. Among these proteins, glutathione peroxidase (ID: Q6C7A8) and superoxide dismutase (ID: Q6C662) were found to be overexpressed. These proteins confer tolerance to the oxidative stress imposed by Cu^+2^, which was also observed in *Candida albicans* [58] and *S. cerevisiae* [59]. It is known that Cu^+2^ ions maintain the activity of cytochrome oxidase, giving yeasts the ability to grow in various culture media with nonfermentable carbon sources [59]. In *Y. lipolytica* AMJ6, the synthesis of cytochrome C oxidase and NADH-cytochrome reductase subunits was positively affected in the presence of Cu^+2^. Our results also showed an increase in the production of proteins that participate in the synthesis of ATP and ion transport (H^+^) under Cu^+2^ treatment. Based on these results, it was not possible to determine whether these proteins were from mitochondria or vacuoles. It is known that the subunits of the H^+^-ATPase vacuolar complex are organized similarly to those of the F1F0-ATPase of mitochondria [60]. Regarding tolerance of Cu^+2^ stress. Eide demonstrated that the H^+^-ATPase vacuolar complex of *S. cerevisiae* participates in cellular detoxification in the presence of this ion [61]. Therefore, it is possible that in *Y. lipolytica* AMJ6, this complex is located in the vacuolar membrane. This suggests that the treatment with Cu^+**2**^ was not harmful to *Y. lipolytica* AMJ6, as a large number of differentially regulated proteins related to protein metabolism, ATP synthesis, and ion transport were identified. Of these upregulated proteins, 4.8% were related to oxidative stress, compared to the other heavy metal ion treatment results of the present study, as were 7.69% of the downregulated proteins.

The proteomic analysis of the *Y. lipolytica* AMJ6 strain in the presence of Cu^+2^ also showed the overproduction of proteins that participate in carbohydrate metabolism for cell wall synthesis, which is related to the resistance of yeasts to various heavy metals in their ionic form. It was reported in [62] that the resistance or tolerance to various concentrations of Cu^+2^ in *S. cerevisiae* is due to the biosorption of the metal in the cell wall. The biosorption capacity of Cu^+2^ through the cell wall has also been observed in several species of mycelial fungi. This feature was described as being the first line of defense of yeasts against excessive levels of Cu^+2^ [63]. Our proteomic results corroborated those of other studies on the Cu^+2^ tolerance of specific yeasts isolated from cold environments. Kan studied the synthesis of diverse proteins in the yeast *Rhodotorula mucilaginosa* AN5, isolated from the sea ice of the Antarctic continent [64]. They observed the overproduction of different groups of proteins with different biological functions, as well as proteins that participate in carbohydrate and energy metabolism, nucleic acid and protein metabolism, protein folding, antioxidant systems, and signaling systems. The synthesis of these types of proteins may contribute to the resistance to high concentrations of not only Cu^+2^, but also other heavy metals.

The number of proteins upregulated in the presence of Cr^+6^, Cd^+2^, and Cu^2+^ was 20, 47 and 112, respectively. Among these proteins, 17 were upregulated in the presence of all three metals individually. The biological function classification of these proteins showed that these proteins participate in the metabolism of other proteins and carbohydrates. We highlight proteins such as peptidases (Q6C1A8), aspartyl protease (Q6C080), and intracellular protease/amidase (Q6C3D2), which have biological activities related to the catabolism of other proteins. Similar results were reported by Hosiner [10], who detected enzymes with proteolytic activity or involved in protein synthesis in *S. cerevisiae* in the presence of As^+3^, Cd^+2^, and Hg^+2^. Proteins involved in carbohydrate metabolism, such as malate dehydrogenase (Q6C8V3) and phosphoglycerate mutase (Q6CFX7), were also identified in the fungus *R. mucilaginosa* AN5 upon treatment with Cd^+2^, Cu^+2^, and Cr^+6^ [64]. Interestingly, enzymes related to oxidative stress, such as peroxiredoxin (Q6CEJ7), thioredoxin reductase (Q6C7L4), and catalase (Q6C3J7), were upregulated in the presence of these heavy metal ions, which was also reported in filamentous fungi [65] and yeasts [66].

In the presence of Cd^+2^ ions, upregulated proteins, such as malate dehydrogenase (Q6C8V3) and glycoside hydrolase (Q6C6P1), enzymes that participate in the synthesis of carbohydrates, showed increased abundance. However, under exposure to Cu^+2^ ions, the Vps5p protein (Q6CGR9) and ribosomal S17 subunit protein (Q6C1P8) were identified, which participate in the transport and recovery of vacuolar proteins and in the general metabolism of proteins, respectively. This variation in the upregulated proteins showed that the yeast AMJ6 activates its cellular machinery by regulating and/or activating its metabolism depending on the metal ion it is exposed to. Figure 14 shows the difference in protein expression against the metal ions evaluated (Cr^+6^, Cd^+2^, and Cu^+2^).

An important group of common proteins that were upregulated under exposure to the three metal ions were related to oxidative stress processes, such as thioredoxin reductase (Q6C7L4), peroxiredoxin TSA1 (Q6CEJ7), and NADPH-FMN oxidoreductase (Q6CFJ5). Exposure to Cd^+2^ ions induced greater upregulation of the last protein (Q6CFJ5) compared to exposure to the other metal ions, followed by peroxiredoxin TSA1 (Q6CEJ7) and thioredoxin reductase (Q6C7L4), which were identified under exposure to Cu^+2^ ions; these proteins are related to the oxidative stress response, as shown in the interaction networks and biological function classification of the proteins (Figure 14). We also found upregulated proteins such as NADH-flavin oxidoreductase (Q6CFJ5), a protein that acts in redox balance. It is important to mention that common upregulated proteins associated with nucleic acid synthesis, cell division, membrane synthesis, and the cell wall, and proteins involved in cell growth, were not found (Appendix A).

The 17 proteins presented differential expression in the presence of each of the metal ions. For example, among the proteins upregulated in the presence of Cr^+6^ ions, aldo/keto reductase (Q6C177) was the most abundant; this enzyme participates in oxidation–reduction and carbohydrate metabolism. This was followed by metallopeptidase (Q6C1A8), an enzyme that participates in the general metabolism of proteins, which was identified under treatment with the same ion (Figure 15). The findings of this research indicate that treatments with these three metal ions cause stress in the cell, depending on the toxicity of each metal, activating the cellular machinery to counteract oxidative stress. In *Y. lipolytica* AMJ6 exposed to the three metal ions, seventeen common upregulated proteins related to carbohydrate metabolism and antioxidant activity were found. The interactions of each protein are shown in Figure 14. Studies carried out on *Meyerozyma guilliermondii* by Ruas [67] identified superoxide dismutase (A5DA67), as an “upregulated” protein in the presence of oxidative stress caused by manganese (Mn^+2^), as well as glutathione synthetase (Q6CBL1) and glutathione reductase (Q6C5H4). In the present investigation, upregulation of these proteins was observed in response to oxidative stress caused by metal ions.

## 5. Conclusions

This is the first work to report the isolation and study of the resistance to various heavy metals and production of proteins in the yeast *Y. lipolytica* isolated from mine tailings from high Andean areas of Peru. The *Y. lipolytica* AMJ6 strain was more tolerant to various heavy metal ions, namely Cr^+6^, Cd^+2^, and Cu^+2^, than other fungal genera previously reported. The proteomic analysis carried out under treatments with these metals showed an abundance change upon treatments with Cu^+2^, followed by Cd^+2^ and Cr^+6^. In all the treatments, the proteins found to be up- and downregulated were proteins related to oxidative stress, such as thioredoxin reductase (Q6C7L4), peroxiredoxin TSA1 (Q6CEJ7), NADPH-FMN oxidoreductase (Q6CFJ5), superoxide dismutase, and catalase, suggesting that the yeast may be altering the production of certain proteins to counteract this stress. For upstream proteins that were upregulated, the interaction network showed that the predominant biological processes were related to oxidative stress and carbohydrate metabolism. Exclusively overexpressed proteins were found under each treatment and could be associated with metal metabolism or with a defense and/or survival response, which makes these yeast strains interesting for biotechnological research, especially for research on metal absorption or metabolite production.

## Figures and Tables

**Figure 1 microorganisms-10-02002-f001:**
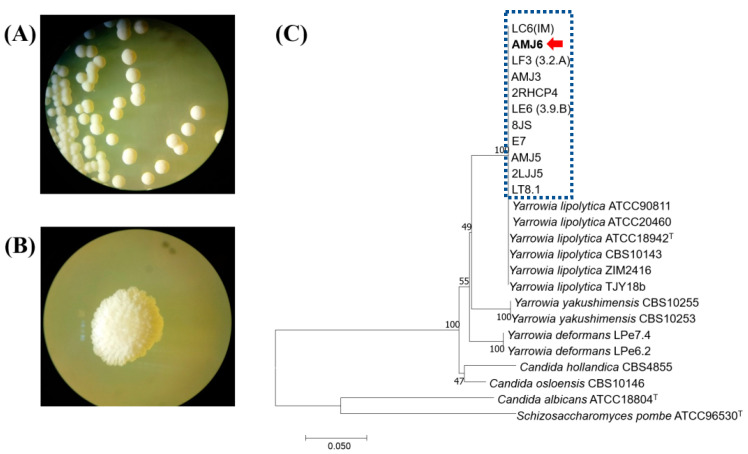
Culture characteristics of the strains (**A**) AMJ3 and (**B**) AMJ6 observed using a stereomicroscope with 10× magnification. (**C**) Phylogenetic tree based on the D1/D2 domain of 28S rDNA of the 10 isolated heavy-metal-resistant strains (blue box), which were phylogenetically related to *Y. lipolytica*. The red arrow indicates the *Y. lipolytica* AMJ6 strain.

**Figure 2 microorganisms-10-02002-f002:**
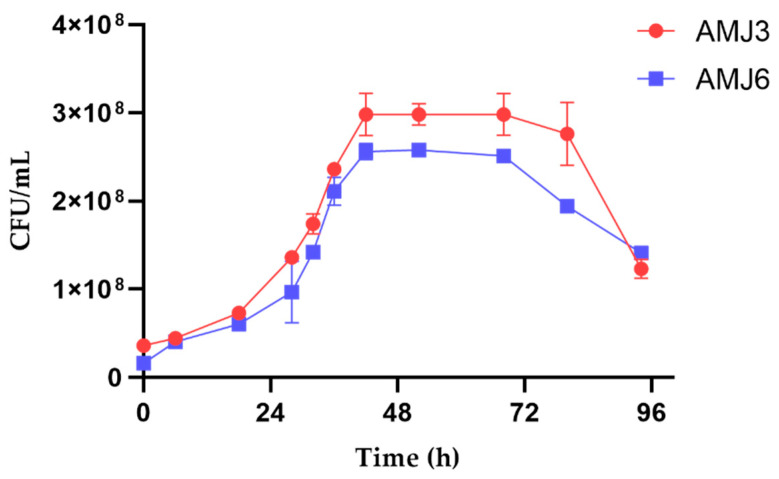
Growth curves of the *Y. lipolytica* AMJ3 and AMJ6 strains in YPG medium, at a temperature of 20 °C and with constant agitation at 150 rpm. The growth was measured over 96 h.

**Figure 3 microorganisms-10-02002-f003:**
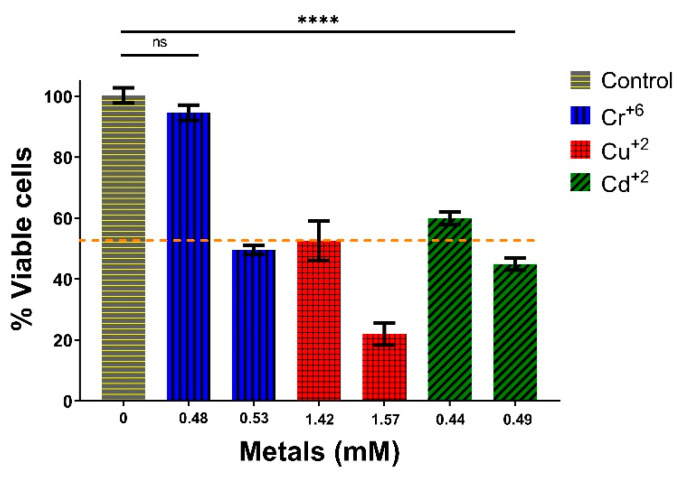
Median lethal concentration (LC_50_) of the heavy metal ions Cr^+6^, Cu^+2^, and Cd^+2^ for *Y. lipolytica* AMJ6 in YPG broth for a period of 3 h. The bars on the graph represent the mean (±SEM) of three independent experiments. The yellow dashed line indicates the concentration of metal at which 50% of the yeast population died. Statistical significance, *p* ≥ 0.05 (Not significant ns) and **** *p* < 0.0001.

**Figure 4 microorganisms-10-02002-f004:**
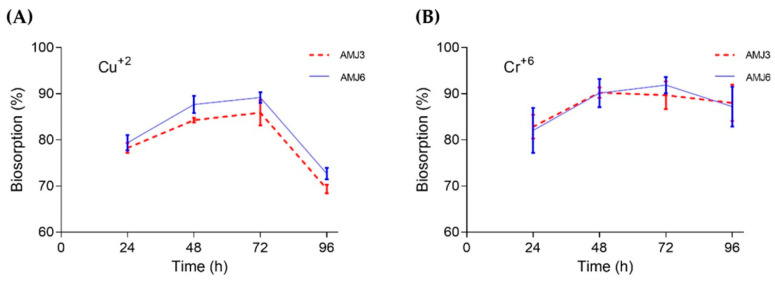
Biosorption of Cu^+2^ (**A**) ions in YPG broth (126 mM), and Cr^+6^ (**B**) ions (0.289 mM) in YPG broth using the *Y. lipolytica* AMJ3 and AMJ6 strains.

**Figure 5 microorganisms-10-02002-f005:**
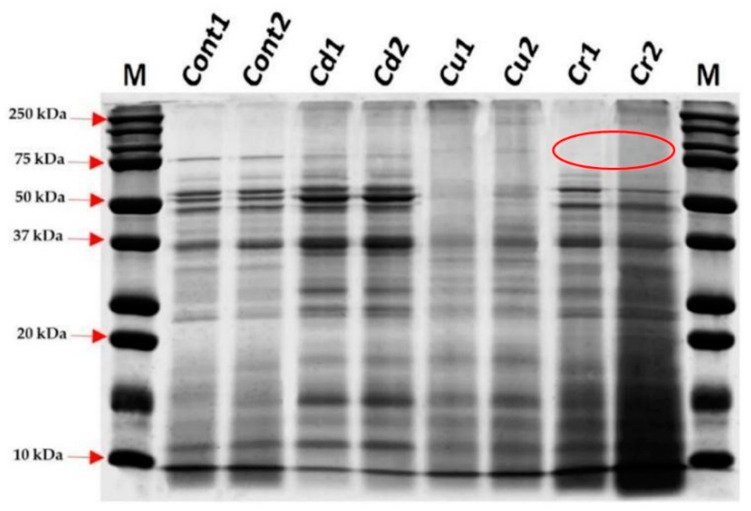
Protein profile of *Y. lipolytica* AMJ6 under treatment with heavy metal ions. Cd1 and Cd2: treatment with Cd^+2^; Cu1 and Cu2: treatment with Cu^+2^; Cr1 and Cr2: treatment with Cr^+6^, Cont1 and Cont2: proteins without treatment (controls). M: Protein ladder (10-260 kDa Thermo Fisher Scientific). Twenty-five micrograms of protein was added to each lane. The samples treated with Cr^+6^ showed an absence of bands of high molecular weight in comparison to the control (red circle).

**Figure 6 microorganisms-10-02002-f006:**
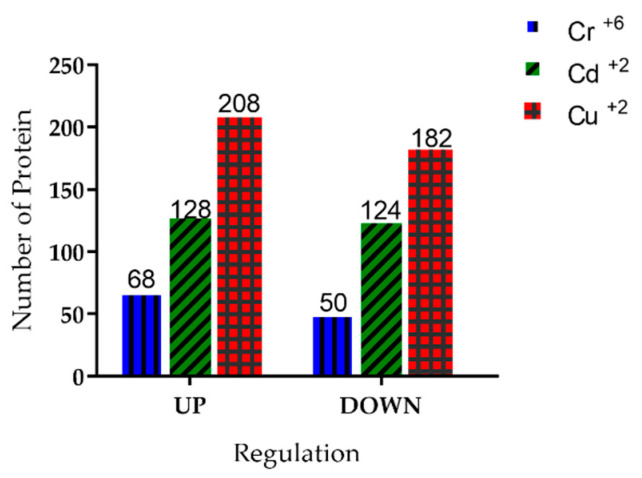
Proteins differentially regulated under treatment with Cr^+6^, Cd^+2^, and Cu^+2^ ions. A total of 118 proteins were differentially regulated in the presence of Cr^6+^, 252 in the presence of Cd^+2^, and 390 in the presence of Cu^+2^ ions. The bars indicate the number of proteins identified under treatment with each metal. The changes in protein expression were relative to the control.

**Figure 7 microorganisms-10-02002-f007:**
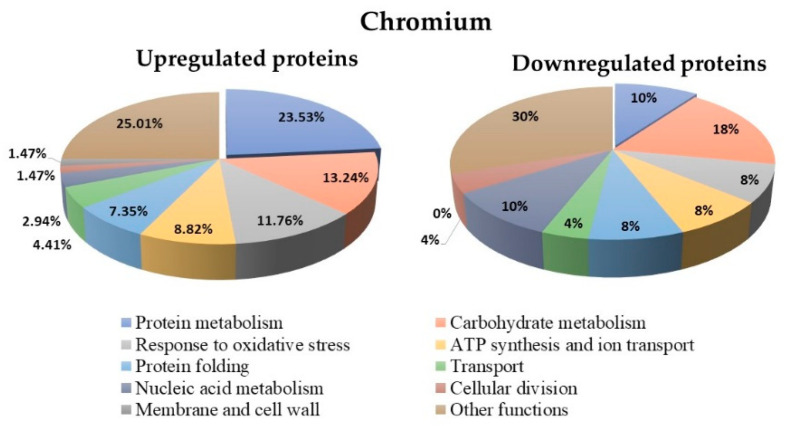
Biological function classification of the 118 up- and downregulated proteins under Cr^2+^ ion treatment in *Y. lipolytica* AMJ6.

**Figure 8 microorganisms-10-02002-f008:**
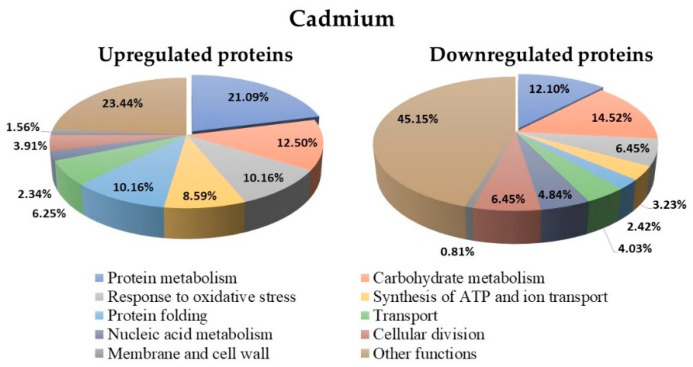
Biological function classification of the 252 differentially regulated proteins under Cd^2+^ ion treatment in *Y. lipolytica* AMJ6. The predominant proteins in both the up- and downregulated groups were related to protein metabolism.

**Figure 9 microorganisms-10-02002-f009:**
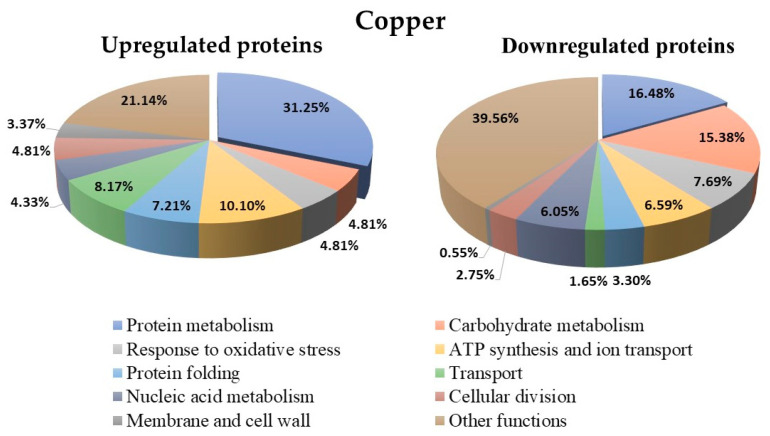
Biological function classification of the 390 differentially expressed proteins under treatment with Cu^+2^ in *Y. lipolytica* AMJ6.

**Figure 10 microorganisms-10-02002-f010:**
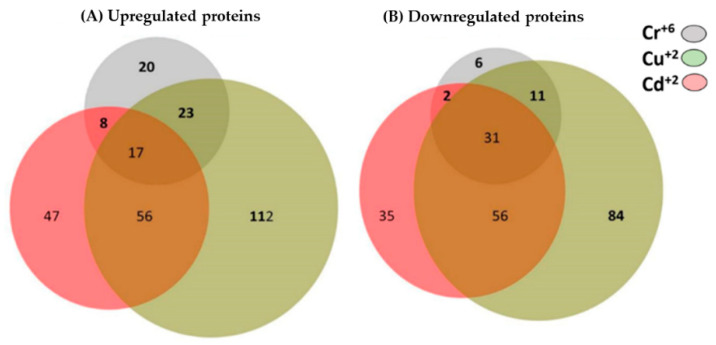
Venn diagram representation of the distribution of upregulated (**A**) and downregulated (**B**) proteins observed under treatment with the heavy metal ions chromium, copper, and cadmium.

**Figure 11 microorganisms-10-02002-f011:**
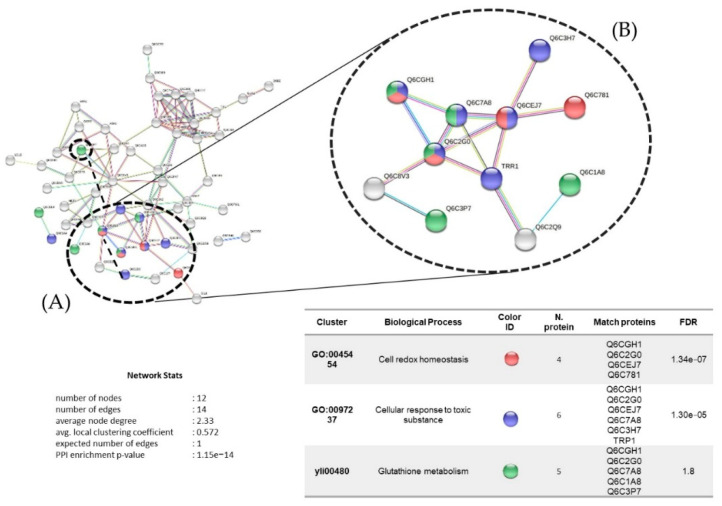
(**A**) Protein interaction networks (upregulated) of *Y. lipolytica* AMJ6 related to biological processes under chromium (Cr^+6^) treatment. (**B**) Proteins related to cell redox homeostasis, cellular response to toxic substances, and glutathione metabolism.

**Figure 12 microorganisms-10-02002-f012:**
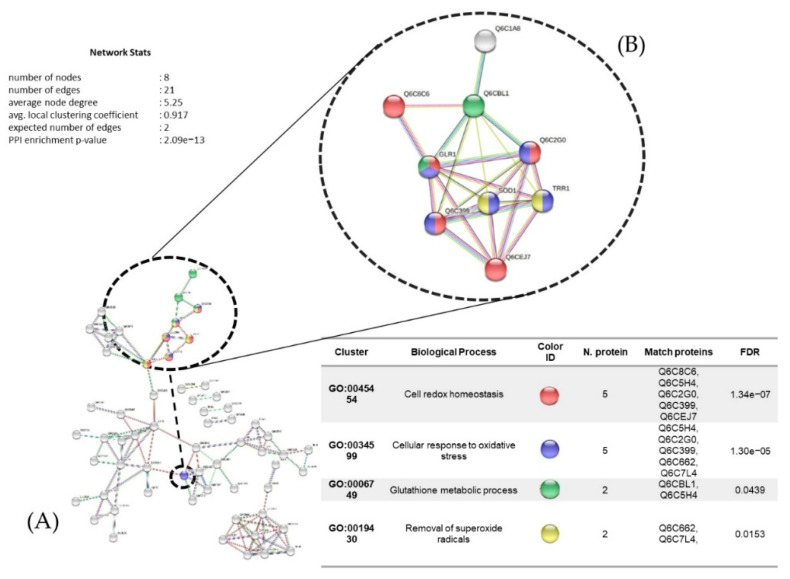
(**A**) Protein interaction networks (upregulated) of *Y. lipolytica* AMJ6 related to biological processes in the copper (Cu^+2^) treatment. (**B**) Proteins related to the response to oxidative stress, cellular response to chemical stimulus, and response to ROS.

**Figure 13 microorganisms-10-02002-f013:**
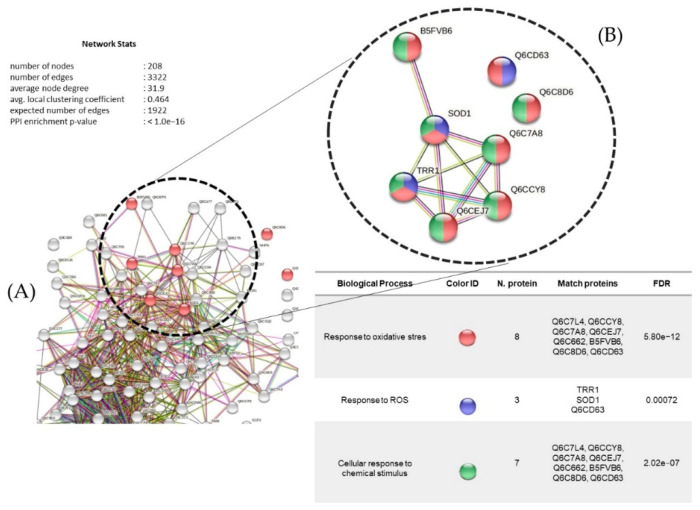
(**A**) Protein interaction networks (upregulated) of *Y. lipolytica* AMJ6 related to biological processes under copper (Cu^+2^) treatment. (**B**) Proteins related to the response to oxidative stress, cellular response to chemical stimulus, and response to ROS.

**Figure 14 microorganisms-10-02002-f014:**
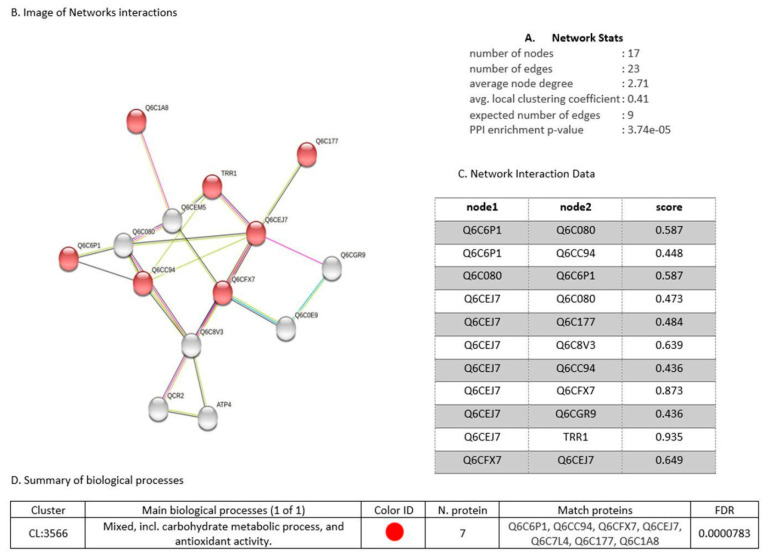
(**A**) Interaction statistics points of 17 common proteins (**B**) Image of the networking relation common proteins (**C**) The points score connection (node) of the networking common proteins showed (**D**) upregulated proteins, related to carbohydrate metabolism and antioxidant activity.

**Figure 15 microorganisms-10-02002-f015:**
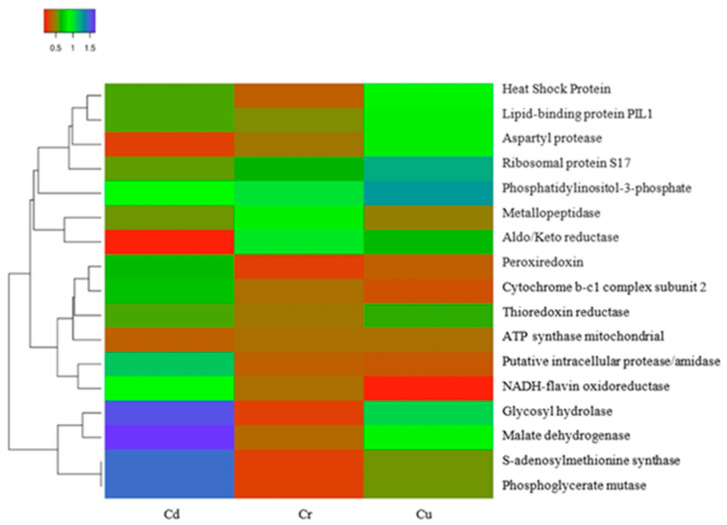
Heatmap presenting most highly differentially expressed proteins in the presence of heavy metals. Proteomic profiling was performed in the presence of metal ions, namely Cd, Cr, and Cu, shown in the columns.

**Table 1 microorganisms-10-02002-t001:** Sampling stations and averages of physicochemical parameters of the samples collected from Lake Junín and the Yanamate tailings.

Sampling Month	Sampling Station	Coordinates UTM 18L		Altitude	pH	Conductivity	TDS ^1^	Temperature
		East	South	(m.a.s.l.)		(µS/cm)	(ppt)	(°C)
June 2015	M1 ^a^	363540.5	8814894.33	4346	2.58	4040	2.12	10.7
	M2 ^a^	363824.88	8815541.09	4345	2.18	12,320	6.27	13.6
	M3 ^a^	363864.47	8815626.03	4345	3.32	9200	1.4	12.5
	M4 ^b^	378916.314	8772988.85	4088	7.21	320	0.16	12.7
	M5 ^b^	374859.529	8778313.75	4084	7.49	1050	0.52	11
	M6 ^b^	369824.633	8781884.24	4084	7.9	430	0.21	13.5
	M7 ^b^	363858.889	8790283.59	4083	7.72	260	0.13	13
November 2015	M1 ^a^	363540.5	8814894.33	4346	1.74	3999	2	10
	M2 ^a^	363824.88	8815541.09	4345	1.71	3999	2	10.5
	M3 ^a^	363864.47	8815626.03	4345	1.73	2173	1.085	11
	M4 ^b^	378916.314	8772988.85	4088	7.29	200	0.08	8
	M5 ^b^	374859.529	8778313.75	4084	7.14	500	0.15	11.9
	M6 ^b^	369824.633	8781884.24	4084	7.24	160	0.34	11
	M7 ^b^	363858.889	8790283.59	4083	7.33	400	0.18	14

^1^ TDS: Total dissolved solids, ^a^ Yanamate, ^b^ Junín.

**Table 2 microorganisms-10-02002-t002:** Heavy metal concentrations in water samples from Junín Lake.

Metals	Lake Junín—Sampling Station M 5		* Maximum Permissible Limit ^1^ (µM)
	June 2015	November 2015	
Arsenic	2.71	2.44	2
Cadmium	0.41	0.14	0.0022
Copper	66.1	11.02	1.57
Chromium	0.21	0.21	0.21
Nickel	1.06	0.39	0.88
Lead	7.72	7.72	0.01
Selenium	<0.12	<0.12	0.06

^1^ The values that exceeded the maximum permissible limits are shown in the grey cells. * Maximum permissible limit (MPL) for Category 4: Conservation of the aquatic environment. (ECA, 2017. Supreme Decree No. 004-2017-MINAM, Peru).

## Data Availability

The datasets presented in this study can be found in online repositories. https://www.ncbi.nlm.nih.gov/summited (accessed on 20 July 2018) with accession number ID:MH656398 and MH656399.

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
