# Peer review of "Proteomic Study of Response to Copper, Cadmium, and Chrome Ion Stress in Yarrowia lipolytica Strains Isolated from Andean Mine Tailings in Peru"

_microorganisms, 2022, doi:10.3390/microorganisms10102002_

Round 1

Reviewer 1 Report

The manuscript entitled “Proteomic study in Yarrowia lipolytica strains isolated from 2 Andean mine tailings from Peru in response to copper, cadmium and chrome ions stress” analyzed the proteins synthesized by the yeast Yarrowia lipolytica AMJ6 (Yl-18 AMJ6) isolated from Andean mine tailings in Peru. The extremophilic YI-18 AMJ6 is resistant to multiple heavy metal ions, which presents LC50 of 1.06 mM, 1.42 mM, 0.49 mM for the Cr+6, Cu+2 and Cd+2 ions, respectively. 

Further, the proteomic analysis of Yl-AMJ6 strain under heavy metal stress showed that several proteins were up- and down-regulated, the biological function of which are involved in the metabolism of macromolecules (proteins, nucleic acids, carbohydrates). More importantly, the highest protein changes were related to oxidative stress and carbohydrate metabolism, indicating a defense mechanism of these yeasts to resist the impact of the environment contaminated by heavy metal ions.

The findings of the study are important to uncover the mechanisms of tolerance to the heavy metal ions. I would suggest the following before the manuscript is suitable for publication:

1. The author should show the unit of heavy metal concentration in Table 2. Is that ppm? 

2. Extensive english grammar, consistent formatting and spelling revisions: 

I.e. Line 273, 274, and 276, “Y. lipolytica” should be italic font.

3. Figure 6 y-axis: it should be “ No. of proteins” or “Number of proteins”

4. The author should clearly define the significance for up- and down-regulation of protein. For example, if 2 fold-changes of protein are significant. Based on the significance, how many proteins significantly change the expression in figure 6

Author Response

Dear Reviewer,

I appreciate your comments and observations on the submitted manuscript. I think they are relevant to improve the article.

Our manuscript has been reviewed by American Journal Expert (verification code 0142-07DD-C8AA-E049-44FP see: https://secure.aje.com/en/certificate/verify) to improve the grammar English.

All changes are in blue (attachment: microorganisms-1898426 edit)

Point 1: The author should show the unit of heavy metal concentration in Table 2. Is that ppm? 

Response 1: Thank you for your analysis comments and suggestions.

Table 2 was modified to μM concentration.

Point 2: Extensive english grammar, consistent formatting and spelling revisions: I.e. Line 273, 274, and 276, “Y. lipolytica” should be italic font.

Response 2: Thank you for your observation.

The scientific names were revised, and changes were made to the blue colour.

Point 3: Figure 6 y-axis: it should be “ No. of proteins” or “Number of proteins”

Response 3: Thank you for your observation.

The Figure 6, the y-axis was modified to “Number of proteins”

Point 4: The author should clearly define the significance for up- and down-regulation of protein. For example, if 2-fold-changes of protein are significant. Based on the significance, how many proteins significantly change the expression in figure 6.

Response 4: Thank you for your observation.

The results of the treatments with the different metals were normalized with the control without treatment (Supplementary Figure S1). (Add Line 230 - 232), Figure 6 shows the regulated proteins, which were significant and normalized with the control without treatment.

Reviewer 2 Report

A very interesting work.

To begin with, I would like to make some general clarifications, because it was quite difficult for me to follow the results and the discussions:

 What other microorganisms were found in these samples?

Introduction

It is mentioned that the concentration of metals has increased recently.. what is this concentration? What is the cause of these increases? These aspects should be introduced in the work

 Why were only the 3 Metals studied? What deposits are there in the area?

L 85 refers to table 1 and table 2. ...if we analyze sentences a and the table have no connection!

Several acronyms are found in the work, but some do not have full names specified (eg L86? ANA?)

How many such tests have been carried out? only in L 134, 145, 164, this was specified" all tests were performed in triplicate". how many copies of the rest of the tests were made?

AMJ3 is first encountered at L 131. What is the difference between AMJ 3 and AMJ 6?

 Is it necessary to show the pH value every time in parentheses? "YPG broth (pH 4.0)"

 In some areas/sections (eg 2.11 and 2.12) too many data are presented that are known to scientists in this field, and subsection 2.13 actually represents the purpose of the study.

L 248: it appears that the samples were collected from Lake Junín and the Yanamate tailings. This does not appear in section 2.2 either. I think it should be clearly specified what type of samples they are and where exactly they are taken from (that is, the same name should be used everywhere. in the abstract it speaks of waste, in section 2 of lake and then in table 2 both names are specified) it creates confusion.

Table 2. Why are 7 metals shown? Since the study is based on 3 metals? What was the logic regarding the choice of the 3 metals studied in the paper? Why are only values ​​from point M5 shown? It would be good to mention their choice

FIGURE 2:

everywhere in the article CFU/mL is used, but in figure 2 CFU mL-1 appears

lipolytica AMJ3 (- -) and AMJ6 (--) IS it necessary to specify the symbols when naming the figure? These are presented in the legend and have different colors

I suggest using the same time interval as in figure 4: 24h, 48h, 72h, 96h

 Figure 3: the name specifies what the lines on the graph represent. This can be put in the legend. What do the values ​​next to metals represent (eg cr 0.96?). from the graph it is not clear which column is for AMJ 3, respectively AMJ 6

 In figure 4, ppm is used. There is an inconsistency in the work. I recommend using the same measurement units everywhere, because the reader is forced to repeat the work in order to check these values...ppm or mM? possibly a table can be made with these parameters so that it is easier to follow these parameters when they are used in experiments

L 317-319 should not be part of the figure name. this will be inserted in the text

 figure 6: I recommend renaming it. I return to the fact that it is not necessary to explain what is represented.

Starting with section 3.7, only AMJ 6 is discussed. why are the values from AMJ3 not shown? Possibly show a graph with its values and make a statement that they are no longer of interest for the study...

Discussions

L 446-452 too few studies that mention this usefulness and the importance of microorganisms. For example, the following paper Extraction of Metals from Polluted Soils by Bioleaching in Relation to Environmental Risk Assessment, Materials 2022, 15, 3973 can be inserted. https://doi.org/10.3390/ma15113973

The discussion section is very extensive, but difficult to follow considering that no reference is made to the previously presented figures. At the end of the discussions, it should be highlighted what is the selection order of the tracked metals.

Table 1 shows the physico-chemical properties of the samples taken, but apart from the pH no comparison is presented or if they influence the analyzes

Conclusions: the purpose of the studies and their importance should be highlighted better.

 References: the sources are quite old. There is only one citation from 2021, 5 from 2019, but most are from 2005-2006. It is somewhat understandable given that the samples were also taken in 2015, but I think that when the comparison is made with other authors, it is desirable to highlight the least researchers.

Author Response

I appreciate your comments and observations on the submitted manuscript. I think they are relevant to improve the article.

Our manuscript has been reviewed by American Journal Expert (verification code 0142-07DD-C8AA-E049-44FP see: https://secure.aje.com/en/certificate/verify) to improve the grammar English.

All changes are in blue (attachment: microorganisms-1898426 edit)

Point 1: What other microorganisms were found in these samples?

Response 1: Thank you for your analysis comments and suggestions.

We found the following microorganisms described in the following table, some of these yeasts have biotechnological potential as plant growth promoters, and phosphate solubilizers among other properties which we are studying

Table: Yeast found in the analysed sample 

Isolate

Yeast

Place

ID Genbank

1

2LJJ2

Rhodotorula toruloides

Junín

MH656364

2

2LJJ4

Pichia fermentans

Junín

MH656370

3

2LJJ5

Rhodotorula mucilaginosa

Junín

MH656362

4

2LJJ6

Yarrowia lipolytica

Junín

MH656368

5

2SNP2

Meyerozyma guilliermondii

Junín

MH656365

6

2LJJ1

Rhodotorula mucilaginosa

Junín

MH656393

7

AMJ6

Yarrowia lipolytica

Junín

MH656398

8

AMJ5

Yarrowia lipolytica

Junín

MH656374

9

AMJ4

Pichia fermentans

Junín

MH656375

10

AMJ3

Yarrowia lipolytica

Junín

MH656399

11

IM

Yarrowia lipolytica

Junín

MG018377

12

LT81

Yarrowia lipolytica

Junín

MG018364

13

9JS

Yarrowia lipolytica

Junín

MH656381

14

8JS

Yarrowia lipolytica

Junín

MG018378

15

2RYP3

Rhodotorula toruloides

 Yanamate

MH656372

16

2RYP8

Pichia fermentans

 Yanamate

MH656377

17

2RYP9

Rhodotorula toruloides

 Yanamate

MG018358

18

2RMCP0

Pichia sp

 Yanamate

MG018375

19

4LYP0

Meyerozyma guilliermondii

 Yanamate

MH656384

20

4LYP1

Yarrowia lipolytica

 Yanamate

MH656400

21

4RYP2

Yarrowia lipolytica

 Yanamate

MH656395

22

4LYP3

Meyerozyma guilliermondii

 Yanamate

MH656392

23

4LYP4

Coniochaeta fodinicola

 Yanamate

MH656394

Point 2: Introduction

It is mentioned that the concentration of metals has increased recently..

what is this concentration? What is the cause of these increases? These aspects should be introduced in the work

Response 2: Thank you for your comments and suggestions.

Line 43-48 was corrected:

“These polluting heavy metals are discarded into the environment entering rivers, lakes and other bodies of water where there is anthropological activity, such as industrial and mining activity. The elimination of these wastes that contain heavy metals is a serious environmental concern with consequences for public health since these metals are toxic and can be lethal to humans [7,8]”

Point 3: Why were only the 3 Metals studied? What deposits are there in the area?

Response 3: Thank you for your observation.

We studied these three metals for being metals with high concentrations found, (Table 2) in addition to being toxic to humans

Point 4: L 85 refers to table 1 and table 2. ...if we analyze sentences a and the table have no connection!

Response 4: Thank you for your observation.

Line 91  was corrected and Table 1 was deleted.

Point 5: Several acronyms are found in the work, but some do not have full names specified (eg L86? ANA?)

Response 5: Thank you for your observation.

Line 92  was corrected and the acronym was added “Autoridad Nacional del Agua -Peru”

Point 6: How many such tests have been carried out? only in L 134, 145, 164, this was specified" all tests were performed in triplicate". how many copies of the rest of the tests were made?

Response 6: Thank you for your comments

The number of experiments performed on the lines is also described: 110 and 191

Point 7: AMJ3 is first encountered at L 131. What is the difference between AMJ 3 and AMJ 6?

Response 7: Thank you for your observation

In the resistance tests to heavy metals, the AMJ3 and AMJ6 strains presented higher levels of tolerance against Cd+2, Cu+2 and Cr+6, being the AMJ6 strain the one that presented higher levels of resistance for the three ions, with values ​​of 150, 400 and 100 mg/L, due to this it was used for the proteomic study

Line 283 - 285 was added “However, a distinct growth pattern was observed when the strains were exposed to different metal concentrations (Supplementary Table S1). The evaluation of the kinetics and growth patterns was relevant for the subsequent LC50 and proteomics analyses”

Point 8: Is it necessary to show the pH value every time in parentheses? "YPG broth (pH 4.0)"

Response 8: Thank you for your observation

The parenthesis (pH 4.0) was removed.

Point 9: In some areas/sections (eg 2.11 and 2.12) too many data are presented that are known to scientists in this field, and subsection 2.13 actually represents the purpose of the study.

Response 9: Thank you for your analysis comments and suggestions.

Sections 2.11 - 2.13 are known, but there are variations depending on the type of sample to be analysed, due to which we decided to place the details of the analysis for this type of strain as described in section 2.12. Subsection 2.11 was reduced and 2.13 was added “to predict protein interactions using Y. lipolytica strain. The network parameters were obtained by default and with a score of 0.7 (high confidence). Graphs were obtained depicting the interactions of proteins, represented by nodes”

Point 10:  L 248: it appears that the samples were collected from Lake Junín and the Yanamate tailings. This does not appear in section 2.2 either. I think it should be clearly specified what type of samples they are and where exactly they are taken from (that is, the same name should be used everywhere. in the abstract it speaks of waste, in section 2 of lake and then in table 2 both names are specified) it creates confusion.

Response 10: Thank you for your observation

Was added a Yanamate, Junín on the sampling station

Point 11: Table 2. Why are 7 metals shown? Since the study is based on 3 metals? What was the logic regarding the choice of the 3 metals studied in the paper? Why are only values ​​from point M5 shown? It would be good to mention their choice

Response 11: Thank you for your analysis comments and suggestions.

Table 2, was modified to μM concentration, in this study we evaluated three metals because the AMJ3 and AMJ6  strains were highly resistant, in addition to being toxic metals for higher organisms. Station M5 was a midpoint of the samplings in Lake Junin.

Point 12: FIGURE 2:

everywhere in the article CFU/mL is used, but in figure 2 CFU mL-1 appears

Response 12: Thank you for your observation

CFU mL-1 was changed to CFU/mL 

Point 13: lipolytica AMJ3 (- ♦ -) and AMJ6 (-■-) IS it necessary to specify the symbols when naming the figure? These are presented in the legend and have different colors. I suggest using the same time interval as in figure 4: 24h, 48h, 72h, 96h

Response 13: Thank you for your comments and suggestions.

Figure 2 was changed.

Point 14: Figure 3: the name specifies what the lines on the graph represent. This can be put in the legend. What do the values ​​next to metals represent (eg cr 0.96?). from the graph it is not clear which column is for AMJ 3, respectively AMJ 6

Reply: Thank you for your comments and suggestions.

Response 14: Figure 3 was changed

Point 15: In figure 4, ppm is used. There is an inconsistency in the work. I recommend using the same measurement units everywhere, because the reader is forced to repeat the work in order to check these values...ppm or mM? possibly a table can be made with these parameters so that it is easier to follow these parameters when they are used in experiments

Response 15: Thank you for your observation and suggestions.

The legend of figure 4 was changed

Point 16:  L 317-319 should not be part of the figure name. this will be inserted in the text figure 6: I recommend renaming it. I return to the fact that it is not necessary to explain what is represented.

Response 16: Thank you for your observation and suggestions.

The legend of figure 6 was changed

Point 17: Starting with section 3.7, only AMJ 6 is discussed. why are the values from AMJ3 not shown? Possibly show a graph with its values and make a statement that they are no longer of interest for the study…

Response 17: Thank you for your observation and suggestions.

See question 7

Point 18: Discussions

446-452 too few studies that mention this usefulness and the importance of microorganisms. For example, the following paper Extraction of Metals from Polluted Soils by Bioleaching in Relation to Environmental Risk Assessment, Materials 2022, 15, 3973 can be inserted. https://doi.org/10.3390/ma15113973

Response 18: Thank you for your suggestions.

Reference was added (ref. 31)

Point 19: The discussion section is very extensive, but difficult to follow considering that no reference is made to the previously presented figures. At the end of the discussions, it should be highlighted what is the selection order of the tracked metals.

Table 1 shows the physico-chemical properties of the samples taken, but apart from the pH no comparison is presented or if they influence the analyzes

Response 19: Thank you for your observation and suggestions.

Table 1 shows the results of some physicochemical parameters measured in these two bodies of water contaminated. The strains isolated were cultivated at different pHs ( from 2 - 6), to determine their optimal growth. The results determined that for all the strains pH 4 was optimal for growth.

Point 20: Conclusions: the purpose of the studies and their importance should be highlighted better.

Response 20: Thank you for your observation and suggestions.

In the conclusions, we show the importance of isolating and studying these microorganisms with biotechnological potential for later use in the biosorption of metals or the search for metabolites. 

Point 21: References: the sources are quite old. There is only one citation from 2021, 5 from 2019, but most are from 2005-2006. It is somewhat understandable given that the samples were also taken in 2015, but I think that when the comparison is made with other authors, it is desirable to highlight the least researchers.

Response 21: Thank you for your observation and suggestions.

we have updated some references, however, the study data of microorganisms isolated from high Andean areas is limited, for this reason, we use references with more than 10 years

Reviewer 3 Report

Major issue

 Unfortunately, the presented work does not contain convincing evidence that the described Yl-AMJ6 strain is distinguished by resistance to the presence of heavy metal ions? The authors claim to have isolated 10 yeast strains (see lines 256-257). Unfortunately, the authors do not show the results of studies showing that these strains differ from each other in terms of resistance to the presence of heavy metal ions. The situation would be clear if the authors showed the results of culturing these 10 strains in the presence of heavy metal ions (the result of such an experiment for one strain is shown in Figure 3), so that it was possible to clearly identify strains distinguished by resistance. Moreover, in such an experiment it would be worth using yeast from a strain not isolated from the tested water samples, but only a reference laboratory strain. If the authors have performed such comparative studies, the results of these studies should be presented, if there are no such results, they should be performed. The uniqueness of the strain Yl-AMJ6 has to be shown.

According to the data in Table 2, among the three tested metal ions: cadmium, chromium and copper, only the concentrations of cadmium and copper ions were significantly above the norm, the concentration of chromium ions was at the acceptable limit. On the other hand, the concentration of lead ions was well above the acceptable standard. It is not clear from the text of the paper why the authors chose the set of metal ions for the research in this and no other way?

The authors planned the proteomics research very badly and the results are very questionable. The proteins were obtained from the culture of the strain Yl-AMJ6 in the presence of 0.5mM Cu (II), 0.11mM Cd (II) and 0.19mM Cr (VI) (see lines 147-148). If these concentrations are related to the approximate LC50 values ​​(lines 286-290) of 1.42 mM mM Cu (II), 0.49 mM Cd (II) and 0.1.06 mM Cr (VI), the experiment was conducted under completely incomparable conditions . The concentrations of metal ions used for the production of proteins were, in relation to the LC50 concentration, respectively 35% for Cu (II), 22% for Cd (II) and 18% for Cr (VI). If we refer this to the data presented in Figure 6, it is easy to notice that the greatest changes are observed for the Cu (II) ion, but its concentration used in the experiment in relation to the LC50 was the highest, and the smallest changes for the Cr (VI) ion, where the concentration was relatively lowest. In order to correctly assess the effect of different metal ions, it was necessary to use the concentration of metal ions as a constant percentage of the LC50.

Mino issue

Lines 256-257

The authors write fairly cautiously that they have identified 10 yeast strains. It is not clear if it was all Y. lipolitica yeast or other yeast species?

Figure 1.

If the symbols on the LC6 (IM) dendrogram ……. LT8.1 corresponds to the strains identified in this work, ie these symbols 11, and the authors say that they identified 10 strains? If these symbols correspond to the strains identified in this work, why was the genetic material sequencing data deposited in the NCBI database for only two strains?

Figure 3

The description of the drawing does not correspond to the data presented there. This is a drawing showing cell survival at different concentrations of metal ions. This is not a drawing showing the LC50.

Supplementary data

1. Spanish metal names are used in the descriptions.

2. Table 3 for ions does not provide the same type of information as Tables 1 and 2. There is no data on the quality of the measurement (peptide count, coverage, etc.).

Author Response

Dear Reviewer,

I appreciate your comments and observations on the submitted manuscript. I think they are relevant to improve the article.

Our manuscript has been reviewed by American Journal Expert (verification code 0142-07DD-C8AA-E049-44FP see: https://secure.aje.com/en/certificate/verify) to improve the grammar English.

All changes are in blue (attachment: microorganisms-1898426 edit)

Point 1: Unfortunately, the presented work does not contain convincing evidence that the described Yl-AMJ6 strain is distinguished by resistance to the presence of heavy metal ions? The authors claim to have isolated 10 yeast strains (see lines 256-257). Unfortunately, the authors do not show the results of studies showing that these strains differ from each other in terms of resistance to the presence of heavy metal ions. The situation would be clear if the authors showed the results of culturing these 10 strains in the presence of heavy metal ions (the result of such an experiment for one strain is shown in Figure 3), so that it was possible to clearly identify strains distinguished by resistance. Moreover, in such an experiment it would be worth using yeast from a strain not isolated from the tested water samples, but only a reference laboratory strain. If the authors have performed such comparative studies, the results of these studies should be presented, if there are no such results, they should be performed. The uniqueness of the strain Yl-AMJ6 has to be shown.

Response 1: Thank you for your analysis, comments and suggestions.

In reference to the observation, we have carried out the resistance tests of the 10 strains in presence of 3 metal ions to determine their resistance pattern, we are presenting them in Supplementary Table 1, in which the AMJ6 strain presented the highest resistance pattern against the metal ions evaluated.

Supplementary Table S1: Growth pattern of Yarrowia lipolytica strains isolated

Strain

ID Genbank

Lake

Metals

Heavy metals (mg/L)

50

100

150

200

250

300

350

400

AMJ3

MH656399

Junín

Cd+2

+

+

-

-

-

-

-

-

Cu+2

++++

+++

+++

+++

+++

-

-

-

Cr+6

+

-

-

-

-

-

-

-

Hg+2

+

-

-

-

-

-

-

-

IM

MG018377

Junín

Cd+2

+

-

-

-

-

-

-

-

Cu+2

+++

+++

+++

++

++

-

-

-

Cr+6

+

-

-

-

-

-

-

-

Hg+2

-

-

-

-

-

-

-

-

AMJ5

MH656374

Junín

Cd+2

+++

+++

++

-

-

-

-

-

Cu+2

+++

-

-

-

-

-

-

-

Cr+6

+

-

-

-

-

-

-

-

Hg+2

-

-

-

-

-

-

-

-

4RYP2

MH656395

Yanamate

Cd+2

+++

++

+

-

-

-

-

-

Cu+2

+++

+++

+++

+++

+++

+

-

-

Cr+6

+

-

-

-

-

-

-

-

Hg+2

-

-

-

-

-

-

-

-

AMJ6

MH656398

Junín

Cd+2

+++

+++

+

-

-

-

-

-

Cu+2

+++

+++

+++

+++

+++

+++

++

+

Cr+6

+

+

-

-

-

-

-

-

Hg+2

+

-

-

-

-

-

-

-

8JS

MG018378

Junín

Cd+2

+++

-

-

-

-

-

-

-

Cu+2

+++

+++

-

-

-

-

-

-

Cr+6

-

-

-

-

-

-

-

-

Hg+2

-

-

-

-

-

-

-

-

2LCL2

MG018361

Junín

Cd+2

++

+

+

-

-

-

-

-

Cu+2

++

+

-

-

-

-

-

-

Cr+6

+

-

-

-

-

-

-

-

Hg+2

-

-

-

-

-

-

-

-

LT81

MG018364

Junín

Cd+2

++

+

+

-

-

-

-

-

Cu+2

+++

++

+

-

-

-

-

-

Cr+6

-

-

-

-

-

-

-

-

Hg+2

-

-

-

-

-

-

-

-

2LJJ5

MH656362

Junín

Cd+2

+++

++

+

+

-

-

-

-

Cu+2

++

+

+

-

-

-

-

-

Cr+6

-

-

-

-

-

-

-

-

Hg+2

-

-

-

-

-

-

-

-

growth not detected (-); less growth(+); moderate growth(++); high growth(+++ /++++)

Line 283 -287 was added “However, a distinct growth pattern was observed when the strains were exposed to different metal concentrations (Supplementary Table S1). The evaluation of the kinetics and growth patterns was relevant for the subsequent LC50 and proteomics analyses”

Point 2: According to the data in Table 2, among the three tested metal ions: cadmium, chromium and copper, only the concentrations of cadmium and copper ions were significantly above the norm, the concentration of chromium ions was at the acceptable limit. On the other hand, the concentration of lead ions was well above the acceptable standard. It is not clear from the text of the paper why the authors chose the set of metal ions for the research in this and no other way?

 Response 2: Thank you for your analysis and observation.

Table 2 was in different units (ppm) and was changed to µM, to better understand

Table 2 was changed

Point 3: The authors planned the proteomics research very badly and the results are very questionable. The proteins were obtained from the culture of the strain Yl-AMJ6 in the presence of 0.5mM Cu (II), 0.11mM Cd (II) and 0.19mM Cr (VI) (see lines 147-148). If these concentrations are related to the approximate LC50 values ​​(lines 286-290) of 1.42 mM mM Cu (II), 0.49 mM Cd (II) and 0.1.06 mM Cr (VI), the experiment was conducted under completely incomparable conditions . The concentrations of metal ions used for the production of proteins were, in relation to the LC50 concentration, respectively 35% for Cu (II), 22% for Cd (II) and 18% for Cr (VI). If we refer this to the data presented in Figure 6, it is easy to notice that the greatest changes are observed for the Cu (II) ion, but its concentration used in the experiment in relation to the LC50 was the highest, and the smallest changes for the Cr (VI) ion, where the concentration was relatively lowest. In order to correctly assess the effect of different metal ions, it was necessary to use the concentration of metal ions as a constant percentage of the LC50.

 Response 3: Thank you for your analysis and observation.

In relation to the concentrations of the metallic ions used, they were worked according to the resistance pattern tests (Table S1) in which it is evident that the Cr+6 ion was the most toxic, followed by the Cd+2 ion, differentiating the Cu+2 ion due to its lower toxicity. The LC50 was another reference test, in which the toxicity results for the first two metals were similar, close to 0.5 mM (Figure 3), while the LC50 results for the Cu+2 ion were 1.42 mM. Our objective of working with lower concentrations, such as the percentages mentioned (35%, 22% and 18%), was to obtain the greatest possible biomass of yeast (pellets), in the confrontations with the three metal ions, to obtain appropriate concentrations of proteins, during the extraction and purification process, that guarantee the proteomic study, both in the regulation and expression of proteins. Values ​​close to the LC50 of the metallic ions did not ensure suitable biomass in the cultures, much less the protein concentration required for the extraction of total proteins in the present study.

Point 4: Lines 256-257

The authors write fairly cautiously that they have identified 10 yeast strains. It is not clear if it was all Y. lipolitica yeast or other yeast species?

Response 4: Thank you for your analysis, comments and suggestions.

We found 10 strains of Y. lipolytica and other yeast described in the following table, some of these yeasts have biotechnological potential as plant growth promoters, phosphate solubilizers among other properties which we are studying

Table: Yeast found in the analysed sample 

Isolate

Yeast

Place

ID Genbank

1

2LJJ2

Rhodotorula toruloides

Junín

MH656364

2

2LJJ4

Pichia fermentans

Junín

MH656370

3

2LJJ5

Rhodotorula mucilaginosa

Junín

MH656362

4

2LJJ6

Yarrowia lipolytica

Junín

MH656368

5

2SNP2

Meyerozyma guilliermondii

Junín

MH656365

6

2LJJ1

Rhodotorula mucilaginosa

Junín

MH656393

7

AMJ6

Yarrowia lipolytica

Junín

MH656398

8

AMJ5

Yarrowia lipolytica

Junín

MH656374

9

AMJ4

Pichia fermentans

Junín

MH656375

10

AMJ3

Yarrowia lipolytica

Junín

MH656399

11

IM

Yarrowia lipolytica

Junín

MG018377

12

LT81

Yarrowia lipolytica

Junín

MG018364

13

9JS

Yarrowia lipolytica

Junín

MH656381

14

8JS

Yarrowia lipolytica

Junín

MG018378

15

2RYP3

Rhodotorula toruloides

 Yanamate

MH656372

16

2RYP8

Pichia fermentans

 Yanamate

MH656377

17

2RYP9

Rhodotorula toruloides

 Yanamate

MG018358

18

2RMCP0

Pichia sp

 Yanamate

MG018375

19

4LYP0

Meyerozyma guilliermondii

 Yanamate

MH656384

20

4LYP1

Yarrowia lipolytica

 Yanamate

MH656400

21

4RYP2

Yarrowia lipolytica

 Yanamate

MH656395

22

4LYP3

Meyerozyma guilliermondii

 Yanamate

MH656392

23

4LYP4

Coniochaeta fodinicola

 Yanamate

MH656394

Point 5: Figure 1.

If the symbols on the LC6 (IM) dendrogram ……. LT8.1 corresponds to the strains identified in this work, ie these symbols 11, and the authors say that they identified 10 strains? If these symbols correspond to the strains identified in this work, why was the genetic material sequencing data deposited in the NCBI database for only two strains?

Response 5: Thank you for your analysis, comments and suggestions.

All the strains found are deposited in the GenBank, as shown in response 4

Point 6: Figure 3

The description of the drawing does not correspond to the data presented there. This is a drawing showing cell survival at different concentrations of metal ions. This is not a drawing showing the LC50

Response 6: Thank you for your analysis, comments and suggestions.

Figure 3 was changed

Point 7: Supplementary data

  1. Spanish metal names are used in the descriptions.

Response 7: Thank you for your observation.

Supplementary data was revised, and the Spanish names were changed

Point 8:

  1. Table 3 for ions does not provide the same type of information as Tables 1 and 2. There is no data on the quality of the measurement (peptide count, coverage, etc.).

Response 7: Thank you for your observation.

Information in table 3 was hidden, but it has been fixed

Round 2

Reviewer 3 Report

I want to thank authors for their work to improve manuscript. It would be very welcome if authors deposit their MS raw data in some open repository like for example PRIDE (PRIDE - Proteomics Identification Database (ebi.ac.uk))